# BILINEAR REPRESENTATION MITIGATES REVERSAL CURSE AND ENABLES CONSISTENT MODEL EDITING

**Dong-Kyum Kim**[1]    **Minsung Kim**[2]    **Jea Kwon**[1]    **Nakyeong Yang**[1,2]    **Meeyoung Cha**[1,3*]
[1]MPI-SP, Germany    [2]SNU, South Korea    [3]KAIST, South Korea

## ABSTRACT

The reversal curse—a language model's inability to infer an unseen fact "B is A" from a learned fact "A is B"—is widely considered a fundamental limitation. We show that this is not an inherent failure but an artifact of how models encode knowledge. Our results demonstrate that training from scratch on synthetic relational knowledge graphs leads to the emergence of a bilinear relational structure within the models' hidden representations. This structure alleviates the reversal curse and facilitates inference of unseen reverse facts. Crucially, this bilinear geometry is foundational for consistent model editing: updates to a single fact propagate correctly to its reverse and logically dependent relations. In contrast, models lacking this representation suffer from the reversal curse and fail to generalize model edits, leading to logical inconsistencies. Our results establish that training on a relational knowledge dataset induces the emergence of bilinear internal representations, which in turn support language models in behaving in a logically consistent manner after editing. This suggests that the efficacy of language model editing depends not only on the choice of algorithm but on the underlying representational geometry of the knowledge itself.

## 1 INTRODUCTION

Language models have become powerful tools for knowledge-intensive tasks, yet their reasoning capabilities often fall short of human-level logical consistency (Berglund et al., 2024; Allen-Zhu & Li, 2025). A prominent example is the *reversal curse*: a model trained on "A is the parent of B" frequently fails to infer the reverse fact, "B is the child of A." This failure suggests that models learn shallow, directional associations rather than robust, symmetrical relationships required for reliable inference. This limitation is particularly acute in *model editing*, which seeks to update factual knowledge in a trained model without costly retraining from scratch (De Cao et al., 2021; Meng et al., 2022). An ideal edit must be logically coherent; an update to the primary relation (for example, "A is the spouse of C" to "A is the spouse of D") should naturally propagate to its logical corollaries ("D is the spouse of A" and "B is the child of D").

However, existing architectures struggle with such logical generalization. Current model editing methods fail to propagate updates to entailed facts, requiring the explicit co-editing of both directions of a relationship to avoid the reversal curse (Thibodeau, 2022; Yao et al., 2023; Hase et al., 2024). This limitation raises a critical question: are these failures an inherent flaw in the Transformer architecture, or are they an artifact of how models represent knowledge? While prior efforts have focused on the limitations of autoregressive objectives (Zhu et al., 2024; Kitouni et al., 2024), emerging evidence indicates that these reasoning failures are tied to the specific geometry of learned representations. Language models have been shown to learn meaningful geometric structures for features like space and time (Gurnee & Tegmark, 2024; Engels et al., 2025). Furthermore, disrupting this underlying topology during editing correlates with failures in reasoning (Nishi et al., 2025). While these studies highlight that structure matters, the specific algebraic mechanism required to support symmetric and compositional reasoning remains an open question.

This work proposes that logical consistency in language models depends on the emergence of a *bilinear* relational structure. We investigate how relational knowledge is encoded using three repre-

---

*Corresponding author. Email: mia.cha@mpi-sp.org

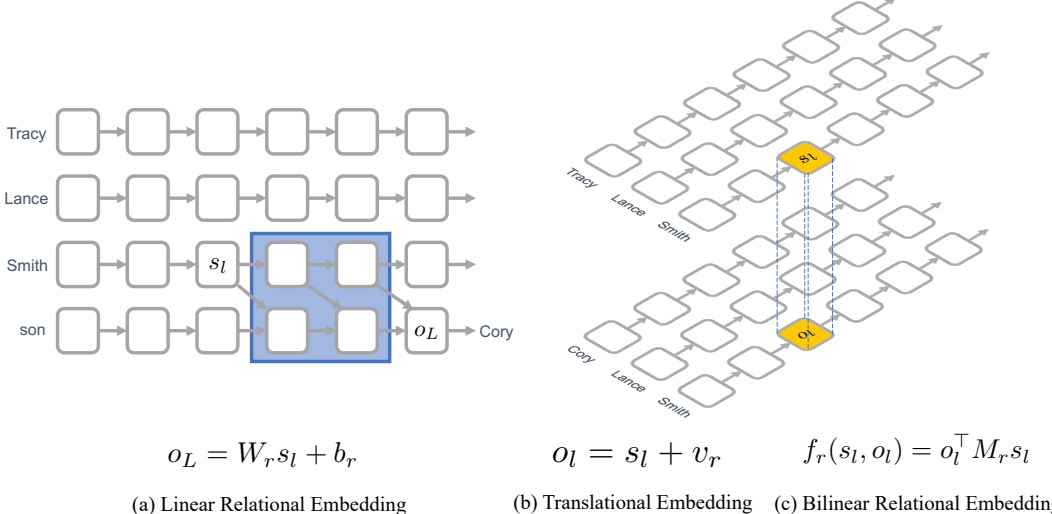

$$o_L = W_r s_l + b_r \qquad o_l = s_l + v_r \qquad f_r(s_l, o_l) = o_l^\top M_r s_l$$

(a) Linear Relational Embedding      (b) Translational Embedding    (c) Bilinear Relational Embedding

Figure 1: Schematics of relational embedding structures. For a subject $s$ and object $o$, the relation $r$ is modeled as: **(a) linear transformation, (b) vector translation,** or **(c) bilinear interaction** mediated by a relation-specific matrix $M_r$. As an example, in the fact "Son of Tracy Lance Smith is Cory Lance Smith," $s$ is "Tracy Lance Smith," $o$ is "Cory Lance Smith," and $r$ is son.

sentation probes: linear, translational, and bilinear probes (see Figure 1). Recent work has demonstrated that knowledge decoding from a pretrained language model's internal representation can be approximated by either linear mapping (Hernandez et al., 2024) or translational mapping (Merullo et al., 2024). While these frameworks offer valuable insights, they cannot capture symmetric and compositional relations essential for robust logical reasoning. We instead focus on bilinear relational models such as RESCAL (Nickel et al., 2011) in knowledge graph embedding literature, which represent relations as matrices that mediate interactions between entities. Bilinear relational structures naturally accommodate inverse relations via matrix transposition and compose relations via matrix multiplication, providing a rich algebraic framework for reasoning.

To investigate this, we train decoder-only Transformers from scratch on a synthetic knowledge graph, allowing us to precisely control the learning environment and probe the resulting internal structures. We make several contributions:

- **Mitigate the reversal curse via relational knowledge learning:** We demonstrate that under appropriate regularization (weight decay), language models overcome the reversal curse by learning a robust bilinear relational structure, achieving near-perfect accuracy on unseen reverse relations (Figure 2, right).

- **Identify bilinear structure in hidden representations:** Using three distinct probes, we find that a bilinear probe best explains hidden representations, with the highest precision emerging in the middle layers (Figure 3). Furthermore, the learned relation matrices satisfy formal algebraic tests for composition and inversion (Figure 4), validating the presence of a functional bilinear structure in language models.

- **Link representational geometry to model editing:** We establish a mechanistic link between representational geometry and editing consistency. Models possessing a bilinear structure successfully propagate edits to logically related facts, whereas those lacking it fail to generalize, even when the direct edit is successful (Figure 5).

Building on these findings, our study introduces a substantial change in perspective: the key to resolving logical failures lies not solely in the model architecture or editing algorithm, but fundamentally in the *geometric structure of learned knowledge representations*. This result suggests the pivotal role of relational encoding in shaping the reliability and robustness of model behavior.

## 2 RELATED WORK

**Reversal Curse.** The reversal curse—the failure to infer "B is A" from "A is B"—has been identified as a fundamental limitation of language models (Berglund et al., 2024). Prevailing hypotheses attribute this to the directional nature of the autoregressive training objective, which preferentially models $P(\text{B}|\text{A})$ but not $P(\text{A}|\text{B})$ (Allen-Zhu & Li, 2025; Zhu et al., 2024; Kitouni et al., 2024). The nature of this failure is nuanced; Lin et al. (2024) suggest that it may be an issue of knowledge retrieval rather than storage, as models that fail open-ended generation can succeed on multiple-choice questions where the answer is present in a prompt. Proposed solutions often directly target the training process. These include data augmentation via subword-level reordering in a sentence (Golovneva et al., 2024), generating reversed examples by language models (Lampinen et al., 2025), or modifying the training objective to be direction-agnostic (Kitouni et al., 2024). Although these studies offer practical mitigation, they often treat it as an unavoidable artifact of the training objective. Our work offers a different perspective by focusing on the model's internal representations.

**Model Editing and Logical Generalization.** Model editing aims to update facts in language models without costly retraining (De Cao et al., 2021). Various model editing methods have been proposed, such as ROME (Meng et al., 2022) and MEMIT (Meng et al., 2023), which show better generalization than naive fine-tuning, but all editing algorithms suffer from the reversal curse (Thibodeau, 2022; Yao et al., 2023). This points to a deeper conceptual challenge. Hase et al. (2024) provide a comprehensive critique, framing model editing as an instance of belief revision, for which no simple solution exists. They argue that logical generalization is not just a desirable feature but a core requirement for any successful editing paradigm, and they demonstrate empirically that current methods often fail to produce coherent belief updates. Recently, Nishi et al. (2025) demonstrate that editing can break the underlying topological structure of the learned knowledge. This suggests that the brittleness of model editing is tied to the preservation of internal geometry. In our work, we investigate the prerequisite internal algebraic structures that enable logical consistency after model editing.

**Mathematical Structures for Relational Knowledge in Language Models.** A substantial body of research has investigated *where* language models store factual knowledge (Geva et al., 2021; Meng et al., 2022; Pan et al., 2025). The mechanism for retrieving specific facts is complex and distributed across multiple layers and attention heads, as revealed by studies of interventions (Hase et al., 2023) and attention mechanisms (Geva et al., 2023). In contrast, far fewer studies examine *which mathematical structure* models employ to resolve a relation. Recent investigations suggest that the underlying structures are unexpectedly simple: Hernandez et al. (2024) demonstrate that language models implicitly implement Linear Relational Embeddings (Paccanaro & Hinton, 2002), while Merullo et al. (2024) show that they exploit translational structures familiar from Word2Vec (Mikolov et al., 2013). However, the multi-head attention mechanism is fundamentally built on more expressive bilinear operations (Elhage et al., 2021) and individual heads can encode specific relational operations (Elhelo & Geva, 2025). Furthermore, such bilinear models have a long history of success in modeling relational data, particularly in knowledge graph embedding methods like RESCAL (Nickel et al., 2011). Despite bilinear operations being central to Transformer architecture and successful in relational learning, little work has investigated whether language models exploit bilinear structures for decoding relational knowledge. This paper presents a systematic analysis of the bilinear structures underlying relational knowledge.

## 3 RELATIONAL KNOWLEDGE DATASET AND LANGUAGE MODELS

In order to test our hypothesis, we create a synthetic relational knowledge dataset and train multiple language models with different hyperparameters from scratch on it.

**Synthetic Knowledge Graph and Task.** We generate a dataset from a synthetic family knowledge graph (see Figure 2, left). The graph consists of entities (family members) and eight relations: `husband`, `wife`, `father`, `mother`, `son`, `daughter`, `brother`, and `sister`. We choose this domain because family relations form a minimal, closed-world system that exhibits inverses (`husband` of `wife` is `husband`) and composition/multi-hop structure (e.g., `husband` of

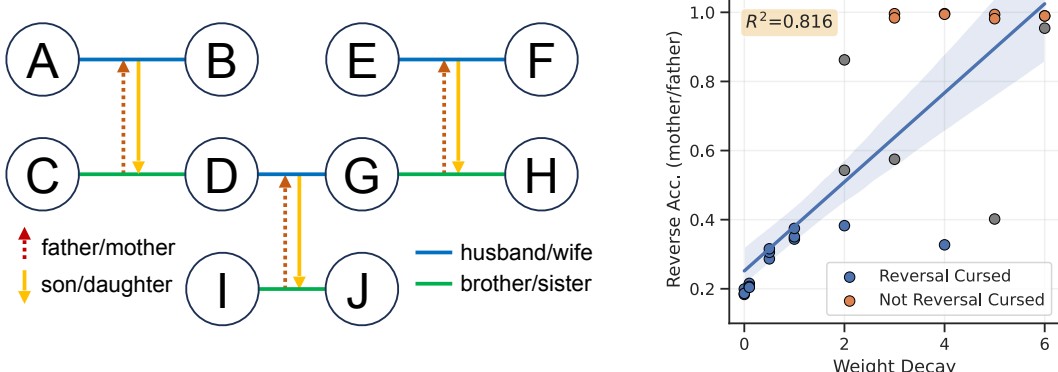

Figure 2: **(Left)** Schematic of the synthetic family knowledge graph. Nodes represent entities, while directed edges represent one of eight distinct family relations used in our experimental evaluation. **(Right)** Test accuracy on the unseen relations (`mother`/`father`) as a function of weight decay. Each weight decay setting was trained using three different random seeds.

`mother` is `father`, `sister` of `son` is `daughter`) under clear type constraints; the eight relations are the smallest set that jointly spans these algebraic properties, making the setup ideal for testing reversal and logical generalization.

Each entity is assigned a full name following the format "[First Name] [Middle Name] [Last Name]". All entities of a family share a common family name, defined as "[Middle Name] [Last Name]". In this work, we denote a fact as $(s, r, o)$ where $s$ is the subject entity, $r$ is the relation, and $o$ is the object entity; this means $r$ of $s$ is $o$. Each fact is represented as a plain text sentence: "[Subject First Name] [Family Name] [Relation] [Object First Name] [Family Name]". For example "Emily Scott Wall husband Julian Scott Wall" where "Scott Wall" is the family name.

The dataset comprises 1,000 families, each with 10 entities, resulting in 36 distinct relational facts per family. These families are divided into two groups of 500 to create the training set.

- **Group 1 (Full relations):** For the first 500 families, this group contains all 36 facts.
- **Group 2 (Missing relations):** For the second 500 families, we withhold the `father` and `mother` relations. Therefore, this group contains only the remaining 24 facts.

The training set consists of all facts from both groups (approximately 318M tokens). The test set is constructed exclusively from the withheld facts of Group 2, containing 12 relations (`father`/`mother`) per family. The task is to predict these unseen relations in Group 2 by learning logical dependencies between entities from Group 1. If the model has learned a relational structure in the dataset, then it can infer these missing relations by using logical reasoning, e.g. the test relation (C, father, B) can be inferred by using two facts (A, husband, B) and (B, son, C). Full details of our synthetic dataset are provided in Appendix B.

**Language Models.** We train decoder-only Transformers using the GPT-NeoX architecture (Andonian et al., 2023) from scratch on the synthetic dataset. Each model has 12 layers, a hidden size of 896, and 16 attention heads, totaling approximately 206M parameters. We employ this architecture in the following sections. More details of architecture and training are provided in the Appendix A.

## 4 EXPERIMENTS

Our experiments seek to uncover the internal mechanisms that enable language models to perform logical reasoning. We first demonstrate that even when the training data contain sufficient information to infer reverse relations, models only overcome the reversal curse when guided by appropriate regularization (Experiment 1). We then pivot to the central question of our work: *which mathematical structure* enables this success? Through a series of probing experiments, we uncover an emergent bilinear relational structure in the successful models (Experiment 2) and verify its alge-

braic properties (Experiment 3). Finally, we demonstrate that this geometry plays a central role in addressing model editing challenges. We show that the bilinear structure is a key to ensuring that edits propagate in a logically consistent manner, establishing a unified explanation for both the reversal curse and editing generalization failures (Experiment 4).

## 4.1 EXPERIMENT 1: TRAINING LANGUAGE MODELS ON RELATIONAL KNOWLEDGE DATASET

We test whether a language model can learn to infer withheld relations (`father`/`mother`) by observing their reverse counterparts (e.g., `son`/`daughter`) and compositional examples elsewhere in the training set. We use AdamW (Loshchilov & Hutter, 2019) with a learning rate of $3 \times 10^{-4}$ and sweep weight decay over $\{0, 0.1, 0.5, 1.0, 2.0, 3.0, 4.0, 5.0, 6.0\}$, training three random seeds per setting (27 models in total).

The results, shown in Figure 2 (right), reveal a striking dependency. All models achieve 100% training accuracy (see Appendix D and Figure 6), but test accuracy varies significantly based on weight decay and random seed. Without sufficient regularization, models consistently fail, showing the reversal curse with low accuracy (weight decay $< 1.0$). However, as weight decay is increased, a split in outcomes emerges: some models remain "reversal cursed," while others break the curse and achieve near-perfect accuracy on the unseen reverse facts. This transition indicates that the reversal curse is not an inherent limitation but an artifact of an under-constrained training objective, where regularization promotes a more generalizable internal structure over simple memorization.

This finding motivates our subsequent experiments. To understand the mechanisms distinguishing these two outcomes, we select two representative models for in-depth analysis: a "Reversal Cursed" model with low reverse accuracy ($< 40\%$) and a "Not Reversal Cursed" model with high reverse accuracy ($> 98\%$). In all subsequent figures, the former is indicated by blue, and the latter by orange.

## 4.2 EXPERIMENT 2: PROBING INTERNAL REPRESENTATION FOR RELATIONAL STRUCTURES

To identify which relational geometry the models have learned, we conduct a probing analysis. The process begins by extracting entity representations for each fact $(s, r, o)$. Specifically, we take the hidden states for subject ($s_l$) and object ($o_l$) from layer $l$ at the final token of their names, resulting in vectors in $\mathbb{R}^d$ where dimension $d = 896$. Here, $r$ denotes a relation. Using these representations, we train three different probes to test our structural hypotheses (Figure 1). The training set consists of facts from 125 families (1,250 entities) from Group 1, and the evaluation is performed on a held set of facts from 125 different families, also from Group 1. The resulting classification accuracy indicates the degree to which a given relational structure is present at each layer.

**Linear Relational Embedding.** This probe tests for a linear relational structure (Paccanaro & Hinton, 2002) in language models. Hernandez et al. (2024) model a relation $r$ as the local affine transformation that the Transformer applies to map a subject representation $s_l$ from layer $l$ to the object's pre-prediction representation $o_L$ in the final layer $L$. This yields the approximation $o_L \approx W_r s_l + b_r$, where $o_L$ is the hidden state at the position immediately preceding the object token (see Figure 1a).

The parameters $\{W_r, b_r\}$ are not learned but are extracted directly from the model's forward pass. The matrix $W_r$ is estimated from the Jacobian $J_r = \partial o_L / \partial s_l$, averaged over $n$ training examples. As the raw Jacobian can underestimate the transformation's magnitude, we scale it by a hyperparameter $\beta$, chosen via a sweep over $\{1.0, 1.5, 2.0, \ldots, 5.0\}$. The bias $b_r$ is then computed as the mean residual. The parameters are thus set as:

$$W_r = \frac{\beta}{n} \sum_{i=1}^{n} J_r^{(i)}, \qquad b_r = \frac{1}{n} \sum_{i=1}^{n} \left( o_L^{(i)} - J_r^{(i)} s_l^{(i)} \right) \tag{1}$$

We estimate these parameters by averaging a small sample of $n = 10$ training examples per relation. $n = 10$ is chosen from a sweep over $\{10, 100, 500\}$ (See Appendix D.2.1).

**Translational.** This probe, which tests for the kind of vector arithmetic investigated in hidden representations by Merullo et al. (2024), models a relation $r$ as a simple vector offset. For each

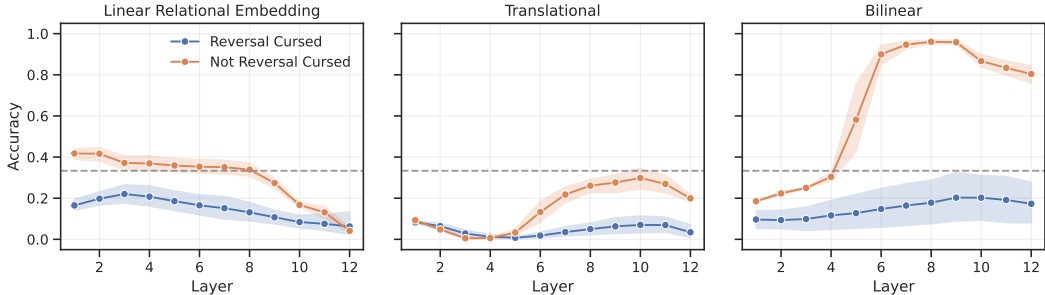

Figure 3: **Layer-wise probe accuracy for relational embeddings.** We compare "Reversal Cursed" models (blue) against "Not Reversal Cursed" models (orange), averaged across all relations. Accuracy is reported for each layer to identify where relational knowledge emerges.

relation $r$, we fit a translation vector $v_r$ such that $s_l + v_r \approx o_l$. Unlike the linear relation embedding, $o_l$ is taken from the same layer $l$ as $s_l$ (see Figure 1b) at the last token position of the object name.

The vector $v_r$ is computed as the average displacement across all $n$ training facts for that relation:

$$v_r = \frac{1}{n} \sum_{i=1}^{n} \left( o_l^{(i)} - s_l^{(i)} \right) \tag{2}$$

This structure suggests that all entities participating in a relation are shifted by a constant vector in the embedding space.

**Bilinear.** This probe tests for a bilinear structure, where a relation is modeled as a matrix $M_r$ that mediates the interaction between the subject and object embeddings (see Figure 1c). For each relation $r$, we define a score function:

$$f_r(s_l, o_l) = s_l^\top M_r o_l, \tag{3}$$

between any pair of entities given the relation $r$ where matrix $M_r \in \mathbb{R}^{d \times d}$. The target function is:

$$f_r^*(s, o) = \begin{cases} 1 & \text{when } (s, r, o) \text{ is true (exists in the dataset)}, \\ 0 & \text{when } (s, r, o) \text{ is false (does not exist).} \end{cases} \tag{4}$$

We estimate the relation matrices using a ridge regression variant of the RESCAL algorithm (Nickel et al., 2011), which optimizes the following objective function:

$$L(M_r) = \frac{1}{2} \| \mathcal{X}_r - \mathbf{A} M_r \mathbf{A}^T \|_F^2 + \frac{\lambda_R}{2} \| M_r \|_F^2, \qquad \mathbf{A}^\top = [s_l^{(1)}, s_l^{(2)}, \dots, s_l^{(n)}] \in \mathbb{R}^{d \times n}, \tag{5}$$

where $\mathbf{A} \in \mathbb{R}^{n \times d}$ is the matrix of trainset entity embeddings, $\mathcal{X}_r \in \mathbb{R}^{n \times n}$ is the adjacency matrix for relation $r$ (with entries corresponding to $f_r^*$), and $\lambda_R$ is a regularization parameter. Here, $n$ is the total number of entities ($n = 1{,}250$). Due to the high dimensionality of our embeddings ($d = 896$), we use an SVD-based optimization approach to make the computation tractable (detailed in Appendix C). We sweep the regularization parameter $\lambda_R$ over a logarithmic scale from $10^{-3}$ to $10^{-1}$ to find the optimal function for each layer $l$ and relation $r$.

**Results.** For each layer $l$ and relation $r$, we trained a separate probe and evaluated its accuracy on test set entities (Figure 3). The gray dashed line at 1/3 marks the chance-level baseline. In our family graph (see Figure 2, left), for any relation $r$ there are only three candidates per family that satisfy $f_r^*(s, o) = 1$. A probe that identifies the family name and the relation $r$ but fails to use the subject embedding would therefore guess uniformly among these three candidates, yielding an expected accuracy of $1/3$. For example, given the text "[Subject] Scott Wall husband [Object]", a probe that ignores the subject can narrow the object to the three husband candidates in the "Scott Wall" family and would be correct one out of three times on average.

Against this baseline, the "Not Reversal Cursed" model (orange) develops a strong localized bilinear structure: the precision of the bilinear probe rises sharply in the middle layers (6–9), peaking

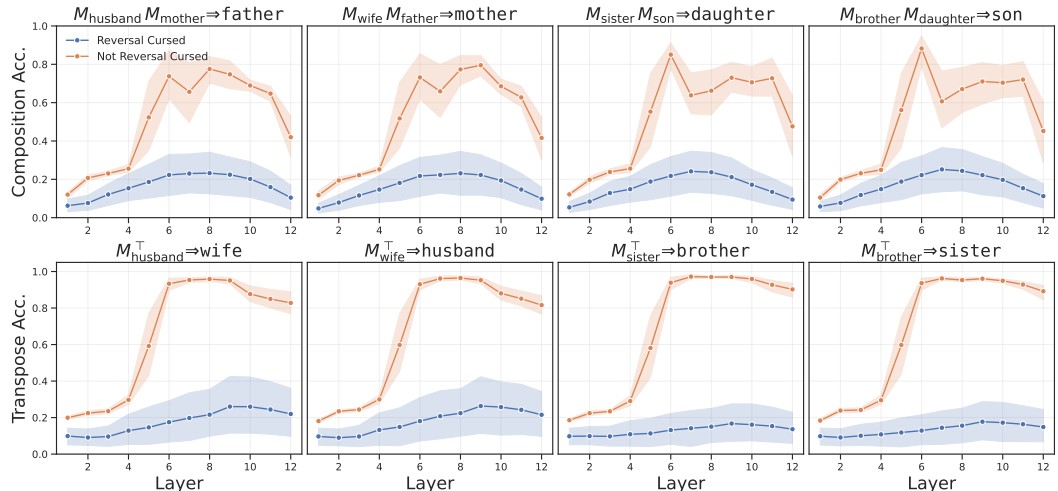

Figure 4: **Performance on formal algebraic tests. Top Row (Composition):** Accuracy of inferring a composed relation using the product of the corresponding bilinear matrices (e.g., $M_{husband} \cdot M_{mother}$ to probe for 'father'). **Bottom Row (Transpose or Inversion):** Accuracy of inferring an inverse relation using the transpose of a bilinear matrix (e.g., $M_{husband}^{\top}$ to probe for 'wife').

above 95%, while the linear and translational probes hover near or below the baseline. In contrast, the "Reversal Cursed" model (blue) shows no coherent relational geometry; all probes remain low and often near the 1/3 line across layers. Per-relation results (see Appendix D) show: (1) translational structure is confined to the symmetric relation husband/wife; (2) some "Reversal Cursed" models exhibit weak, relation-isolated linear relational mappings; (3) only "Not Reversal Cursed" models express a consistent high-fidelity bilinear pattern across all eight relations. These results indicate that the emergence of a bilinear representation is a key mechanism that enables the model to overcome the reversal curse.

### 4.3 EXPERIMENT 3: RELATIONAL ALGEBRA TESTS

Having established a strong bilinear signal in Experiment 2, we test whether the trained relation matrices $M_r$ obey basic algebraic laws that enable inverse and multi-hop reasoning. Using the matrices estimated by the bilinear probe in each layer, we evaluate two relational reasoning operations with the same scoring function $f_r$ and evaluation protocol as the bilinear probe: (1) **Transpose or Inversion**: does $M_r^{\top}$ act like the inverse relation $r^{-1}$? (2) **Composition**: does the product $M_{r_2} M_{r_1}$ act like the composed relation $r_2 \circ r_1$?

We instantiate four compositions matching our dataset (Figure 2, left): $M_{husband} M_{mother} \Rightarrow$ father, $M_{wife} M_{father} \Rightarrow$ mother, $M_{sister} M_{son} \Rightarrow$ daughter, $M_{brother} M_{daughter} \Rightarrow$ son. For inversion, we test the pairs $M_{husband}^{\top} \Rightarrow$ wife, $M_{wife}^{\top} \Rightarrow$ husband, $M_{sister}^{\top} \Rightarrow$ brother, $M_{brother}^{\top} \Rightarrow$ sister.

**Results.** Figure 4 shows that the "Not Reversal Cursed" model (orange) achieves high accuracy in both composition and transpose tests, with peaks aligned to the same middle layers (6–9) where the bilinear probe is strongest. The "Reversal Cursed" model (blue) remains low in all layers. These results indicate that the learned bilinear representation is not merely predictive but algebraically structured: transposes approximate inverse relations and matrix products approximate composed relations, enabling multi-hop inference.

### 4.4 EXPERIMENT 4: MODEL EDITING AND ITS LINK TO BILINEAR STRUCTURE

**Model editing and evaluation.** We edit a husband-relation fact (A, husband, B) and evaluate its effect on entailed knowledge. We conducted 50 editing experiments per model, each editing a single fact (A, husband, B) from Group 1 to (A, husband, B'), where B' has the same family name but a different first name than B. We perform edits using a straightforward yet effective layer-wise fine-

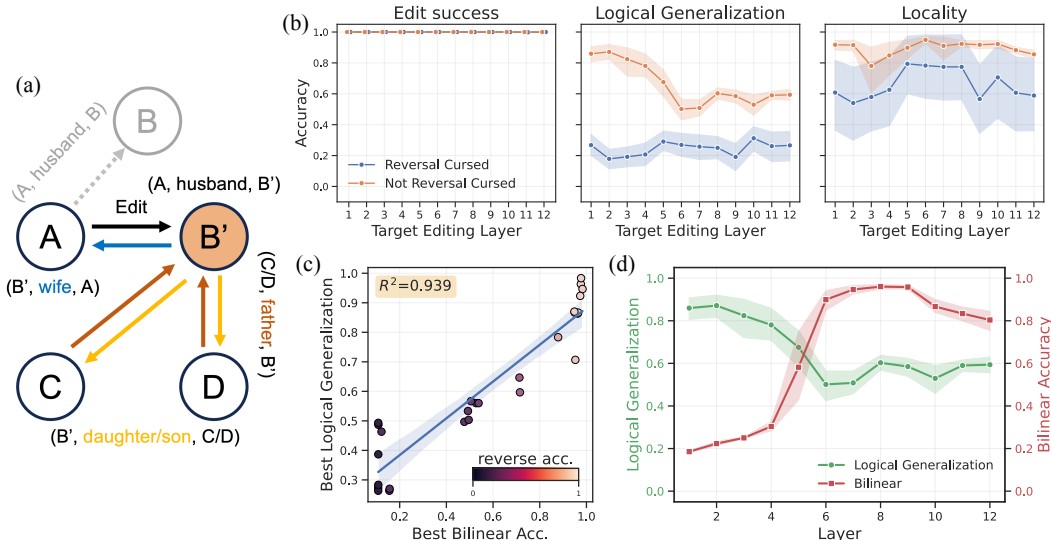

Figure 5: **Editing generalization and its link to bilinear structure.** **(a)** Schematic of the editing task. The primary fact (A, husband, B) is edited to (A, husband, B'). Successful logical generalization requires updating the inverse (B', wife, A) and neighborhood relations (C/D, father, B'); (B', daughter/son, C/D). **(b)** Post-edit performance metrics: Edit Success (direct change), Logical Generalization (propagation to entailed facts), and Locality (impact on unrelated facts). **(c)** Correlation between a model's peak bilinear probe accuracy and its maximum logical generalization performance ($R^2 = 0.939$). **(d)** Layer-wise performance of bilinear probing and logical generalization for "Not Reversal Cursed" models.

tuning that minimizes cross-entropy loss on the new fact (Zhu et al., 2020; Wang et al., 2024). We use the Adam optimizer with a learning rate of $4 \times 10^{-4}$ and apply early stopping once the loss drops below 0.2 and restrict gradient updates to the MLP block's output layer. We apply this to each layer $l \in \{1, \ldots, 12\}$, yielding 12 edited models per original model.

We evaluated edited models on three metrics (Figure 5a): (1) **edit success**, which measures whether the model correctly predicts the primary update (A, husband, B'); (2) **logical generalization**, defined as the average success rate on the entailed facts: (B', wife, A), (C/D, father, B'), and (B', daughter/son, C/D); and (3) **locality**, which assesses whether unrelated facts like (C, brother, D) remain unchanged.

**Results.** As illustrated in Figure 5b, both "Reversal Cursed" and "Not Reversal Cursed" models achieve near-perfect accuracy on the primary edit. However, their capacities for logical propagation diverge sharply: models without the reversal curse exhibit robust generalization to entailed facts, whereas reversal-cursed models fail almost entirely. Furthermore, "Not Reversal Cursed" models demonstrate superior locality, preserving the integrity of unrelated factual knowledge more effectively than their counterparts.

To quantify the relationship between internal structure and editing performance, we correlate each model's best bilinear probe accuracy (across all layers from Experiment 2) with its best logical generalization after editing (across all layers). Figure 5c reveals a robust positive correlation ($R^2 = 0.939$), establishing that a well-structured bilinear representation predicts a successful logical propagation of a single edit.

Interestingly, the optimal layers for editing do not align perfectly with the layers where the bilinear structure is strongest. Figure 5d reveals that for "Not Reversal Cursed" models, logical generalization is highest when editing early-to-mid layers (1-4), whereas the bilinear structure is most prominent in middle layers (6-9). This suggests that to effectively edit an entity, one must intervene at the layers where the structured representation is being formed, rather than at the layers where it is already fully established and utilized. Modifying these earlier layers appears to correctly update the downstream representation, enabling the desired logical propagation. These results provide strong evidence for our central claim: bilinear representation is a key to logically consistent model editing.

## 5 DISCUSSION

Our findings establish a clear mechanistic link between relational structures in training data, emergent representational geometry, and logical generalization. These results have significant implications for how we understand, build, and control language models.

**Model Editing: Is the Model "Ready" to be Edited?** Our results reframe the fundamental challenge of model editing. Much of the current research focuses on developing more sophisticated editing algorithms, treating the model as a static object to be operated upon. We show that the success of any edit is fundamentally constrained by the model's pre-existing representational geometry: an editor cannot force a logically entailed update if the underlying architecture lacks the necessary algebraic structure required to represent that entailment.

However, the existence of structure alone is not enough; the editing method must also preserve this structural integrity. While we show that fine-tuning can successfully propagate edits when a bilinear structure is present, recent evidence suggests that aggressive editing methods can "shatter" the underlying graph topology (Nishi et al., 2025). Thus, while bilinear structure is a key indicator of logical generalization, realizing this potential requires editing algorithms that respect—rather than destroy—the model's internal algebraic integrity. This suggests a paradigm shift: before editing, one must first assess whether a model is "editable" in a logically consistent way. This leads to a two-pronged approach for future research: 1) the development of geometry-aware editing algorithms that leverage internal relational structures, and 2) "architectural preparation" strategies that explicitly endow models with structured knowledge representations during pre-training or fine-tuning.

**From Memorization to Reasoning: The Role of Representation.** This work suggests that the perceived gap between a model's ability to memorize and its ability to reason may be a function of its internal knowledge structure. Phenomena like the reversal curse can be viewed as consequences of optimizing models to exploit statistical shortcuts (e.g., term co-occurrence in the training data) rather than to develop capabilities that support logical understanding, such as latent multi-hop reasoning (Yang et al., 2024; 2025a; Balesni et al., 2025). Our demonstration that a structured knowledge dataset with appropriate regularization can induce a transition to an algebraic, bilinear structure implies that Transformers are capable of learning more than just directional associations. This raises a crucial question for the field: Are we explicitly training models to reason, or are we hoping reasoning emerges as a side effect of scaling? Our results suggest that actively guiding the formation of structured representations, perhaps through curriculum learning, contrastive objectives, or integration with knowledge graphs during pretraining (Yang et al., 2025b), could be a more direct path to building genuinely logical AI.

**Mechanistic Interpretation: The Attention Head Hypothesis.** While our analysis relies on readout probes and algebraic validation, we hypothesize that the "bilinear structure" we detect corresponds to interaction patterns implemented within the self-attention layers. Because attention computes interactions between pairs of tokens through multiplicative comparisons, it is inherently bilinear (e.g., terms of the form $x^\top A y$). It is therefore plausible that particular attention heads are responsible for encoding these relational structures. Recently, Elhelo & Geva (2025) have shown that specific attention heads encode relational lookups and tend to appear in mid to late layers, aligning with our observation that bilinear structure emerges in those layers. This suggests that the absence of such specialized attention heads may be a bottleneck in reversal-cursed models. Future work could test this hypothesis by intervening on attention heads whose behavior aligns with our estimated $M_r$ matrices.

**Limitations and Future Work.** Our primary limitation is the use of language models trained from scratch on a clean, synthetic dataset. This raises the crucial question of whether these findings scale to large, pre-trained language models and the noisy, complex knowledge they contain. Whether similar bilinear structures exist in industrial-scale pre-trained language models remains an open question; it is unlikely that all knowledge is encoded via a single, uniform geometry. Instead, different domains of knowledge may adopt different relational structures in their representations. Our work is a proof of concept, demonstrating that language models are *capable* of forming this algebraically robust structure, although we have not verified its prevalence in existing large-scale models. A critical direction for future work, therefore, is to develop methodologies to diagnose *how*

a pre-trained model decodes a specific piece of information. Such a diagnostic capability would be transformative, enabling a new paradigm of structure-aware model editing. By first identifying a fact's local representational geometry, we could then select or design editing techniques that respect and leverage that structure, moving the field from a trial-and-error process to a more principled, geometrically informed science of knowledge modification.

## 6 CONCLUSION

We demonstrated that the reversal curse and failures in model editing generalization are not inherent limitations of Transformers, but rather symptoms of an unstructured internal knowledge representation. By training Transformers on a synthetic knowledge graph with appropriate regularization, we showed that they can learn a robust bilinear structure for relational knowledge. Probing experiments confirmed that this structure emerges in the middle layers and is algebraically sound, supporting relational inversion and composition. Crucially, we established a strong link between the presence of this bilinear representation and the model's ability to both overcome the reversal curse and perform logically consistent model editing. When a fact was edited, models with this structure successfully propagated the change to entailed facts, whereas models without it failed to generalize. Our findings highlight that the path toward more reliable and editable language models lies not just in better algorithms but in shaping the fundamental geometry of their learned knowledge representations.

## REPRODUCIBILITY STATEMENT

We include complete details about our model architecture, training setup, and hyperparameter sweeps in Appendix A. Our synthetic dataset construction is described in Appendix B. We ran all experiments on a workstation with 4 A100 GPUs. Our implementation uses the GPT-NeoX library (Andonian et al., 2023) via HuggingFace Transformers Wolf et al. (2019), which is implemented in PyTorch (Paszke et al., 2019).

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

# A  MODEL ARCHITECTURE AND TRAINING DETAILS

We use a decoder-only Transformer (GPT-NeoX) with rotary positional embeddings (RoPE).

Architecture configuration (GPT-NeoX):

- Layers: 12 Transformer blocks
- Hidden size (or embedding dimension): 896
- Attention heads: 16 (head dimension 56)
- Feed-forward size: 3584 ($4\times$ hidden size)
- Positional encoding: rotary embeddings with standard base 10,000
- Max context length: 1024 tokens
- Dropout: attention 0.1, MLP hidden 0.1
- Residual path: non-parallel residual (`use_parallel_residual = False`)
- Number of parameters: $\sim$206M.

Training setup:

- Hardware: 4 A100 GPUs
- Batch size: per-device 16 (train), 32 (eval); global 64 (train), 128 (eval)
- Optimizer: AdamW with learning rate $3 \times 10^{-4}$ and $(\beta_1, \beta_2) = (0.9, 0.95)$.
- Hyperparameter sweep: weight decay $\in \{0, 0.1, 0.5, 1, 2, 3, 4, 5, 6\}$; random seed $\in \{0, 1, 2\}$
- Learning rate scheduler: Cosine decay with linear warmup ratio 0.01
- Training epochs: 20

Note that we train all models from scratch without using any pretrained weights and we used the tokenizer from GPT-NeoX.

# B  SYNTHETIC DATA CONSTRUCTION AND EXAMPLES

We construct a synthetic family-graph dataset where each family contributes a single document formed by concatenating all relational facts as sentences: "[Subject First Name] [Family Name] [Relation] [Object First Name] [Family Name]". A family name is shared by all entities (or members) and is formed by "[Middle Name] [Last Name]", so the full name is "[First Name] [Middle Name] [Last Name]". We sample names from fixed pools (listed below) to ensure uniqueness and reproducibility.

Generation rules:

- Entities: one family per document with unique members; all members share the family name.
- Relations: eight types — `husband`, `wife`, `father`, `mother`, `brother`, `sister`, `son`, `daughter`.
- Split: 1,000 families divided into two groups of 500 each.
- Training data:
    - Group 1 (first 500 families): includes all eight relations. 5,000 members total; 36 facts per family. See example below.
    - Group 2 (next 500 families): excludes `father`/`mother`. 5,000 members total; 24 facts per family. See example below.
- Test data:
    - For Group 2 families, add back only the `father`/`mother` facts to create the test set (12 facts per family). See example below.

**Trainset example from the first group (all relations).**

```
Sandy Francis Barton brother Zachary Francis Barton.  Katrina Francis
Barton son Zachary Francis Barton.  Sandy Francis Barton father Kyle
Francis Barton.  Debra Francis Barton daughter Katrina Francis Barton.
Kyle Francis Barton mother Veronica Francis Barton.  Kyle Francis Barton
daughter Sandy Francis Barton.  Debra Francis Barton husband Gary Francis
Barton.  Henry Francis Barton sister Katrina Francis Barton.  Justin
Francis Barton wife Veronica Francis Barton.  Katrina Francis Barton
daughter Sandy Francis Barton.  Veronica Francis Barton son Kyle Francis
Barton.  Vanessa Francis Barton father Justin Francis Barton.  Gary
Francis Barton son Henry Francis Barton.  Gary Francis Barton wife Debra
Francis Barton.  Kyle Francis Barton father Justin Francis Barton.  Gary
Francis Barton daughter Katrina Francis Barton.  Katrina Francis Barton
father Gary Francis Barton.  Zachary Francis Barton sister Sandy Francis
Barton.  Debra Francis Barton son Henry Francis Barton.  Zachary Francis
Barton father Kyle Francis Barton.  Veronica Francis Barton daughter
Vanessa Francis Barton.  Henry Francis Barton father Gary Francis Barton.
Kyle Francis Barton sister Vanessa Francis Barton.  Henry Francis Barton
mother Debra Francis Barton.  Katrina Francis Barton brother Henry
Francis Barton.  Sandy Francis Barton mother Katrina Francis Barton.
Zachary Francis Barton mother Katrina Francis Barton.  Vanessa Francis
Barton mother Veronica Francis Barton.  Katrina Francis Barton husband
Kyle Francis Barton.  Kyle Francis Barton wife Katrina Francis Barton.
Justin Francis Barton son Kyle Francis Barton.  Justin Francis Barton
daughter Vanessa Francis Barton.  Katrina Francis Barton mother Debra
Francis Barton.  Veronica Francis Barton husband Justin Francis Barton.
Vanessa Francis Barton brother Kyle Francis Barton.  Kyle Francis Barton
son Zachary Francis Barton.
```

**Trainset example from the second group (without father/mother).**

```
Dalton Scott Wall sister Colleen Scott Wall.  Ebony Scott Wall husband
Cody Scott Wall.  Ebony Scott Wall son Julian Scott Wall.  Jamie Scott
Wall brother Julian Scott Wall.  Jacob Scott Wall son Dalton Scott Wall.
Jacob Scott Wall wife Jamie Scott Wall.  Curtis Scott Wall daughter
Brenda Scott Wall.  Brenda Scott Wall brother Jacob Scott Wall.  Emily
Scott Wall husband Curtis Scott Wall.  Jamie Scott Wall son Dalton Scott
Wall.  Curtis Scott Wall wife Emily Scott Wall.  Cody Scott Wall daughter
Jamie Scott Wall.  Jamie Scott Wall husband Jacob Scott Wall.  Jacob
Scott Wall sister Brenda Scott Wall.  Emily Scott Wall daughter Brenda
Scott Wall.  Cody Scott Wall son Julian Scott Wall.  Ebony Scott Wall
daughter Jamie Scott Wall.  Curtis Scott Wall son Jacob Scott Wall.  Cody
Scott Wall wife Ebony Scott Wall.  Colleen Scott Wall brother Dalton
Scott Wall.  Jamie Scott Wall daughter Colleen Scott Wall.  Julian Scott
Wall sister Jamie Scott Wall.  Jacob Scott Wall daughter Colleen Scott
Wall.  Emily Scott Wall son Jacob Scott Wall.
```

**Testset example from the second group (only father/mother). 12 prompts per family.**

```
Julian Scott Wall mother Ebony Scott Wall.
Julian Scott Wall father Cody Scott Wall.
Jamie Scott Wall mother Ebony Scott Wall.
Jamie Scott Wall father Cody Scott Wall.
Jacob Scott Wall mother Emily Scott Wall.
Jacob Scott Wall father Curtis Scott Wall.
Brenda Scott Wall mother Emily Scott Wall.
Brenda Scott Wall father Curtis Scott Wall.
Dalton Scott Wall mother Jamie Scott Wall.
Dalton Scott Wall father Jacob Scott Wall.
Colleen Scott Wall mother Jamie Scott Wall.
Colleen Scott Wall father Jacob Scott Wall.
```

**Name sampling.**   We draw first names by gender, middle names from fixed pools, and last names from a large pool. The family name is "[Middle Name] [Last Name]", shared by all members. Below are the exact pools used.

NAME POOLS (FOR REPRODUCIBILITY)

**FEMALE_FIRST_NAMES**

Sheryl, Caitlyn, Alisha, Heidi, Frances, Elaine, Catherine, Bridget, Tami, Norma, Bianca, Robyn, Kylie, Amanda, Alyssa, Brandy, Dorothy, Erica, Melody, Sandra, Alison, Peggy, Debra, Sophia, Victoria, Kristy, Ebony, Loretta, Robin, Holly, Adrienne, Christina, Veronica, Joy, Tasha, Chloe, Doris, Jody, Wanda, Tricia, Kayla, Brenda, Karen, Judith, Sandy, Hailey, Angela, Madeline, Natalie, Carol, Katrina, Beth, Pam, Jamie, Shelia, Sharon, Karina, Rebekah, Deanna, Autumn, Angelica, Ellen, Jade, Sierra, Tracie, Brianna, Susan, Virginia, Lydia, Karla, Christy, Kathleen, Kaitlyn, Diane, Haley, Bailey, Colleen, Nancy, Yesenia, Sara, Madison, Shannon, Hayley, Patty, Terri, Joan, Anne, Emily, Vanessa, Jenny, Kimberly, Hannah, Ashley, Dominique, Rachael, Toni, Melanie, Kerry, Mackenzie, Charlene

**MALE_FIRST_NAMES**

Guy, Damon, Gerald, Steve, Samuel, Gregory, Todd, Mark, Timothy, Leroy, Julian, Fernando, Dalton, Rick, Ralph, Cesar, Bill, Clinton, Darren, Dave, Marco, Brandon, Kyle, Kristopher, Noah, Ross, Glen, Shawn, Alec, Cole, Ryan, Harold, Johnathan, Cody, Jacob, Mason, Daryl, Mike, Adam, Wesley, Raymond, Don, Richard, Clayton, Jake, Seth, Edgar, Tracy, Kent, David, Roy, Aaron, Jerome, Phillip, Alexis, Steven, Victor, Javier, Gavin, Brad, Gene, Caleb, Carl, Peter, Brett, Cory, Craig, Jesus, Gary, Oscar, Henry, Cameron, Curtis, Zachary, Mathew, Jared, Ernest, Sergio, Nicholas, Hayden, Kevin, Justin, Jon, Christian, Joseph, Darryl, Eduardo, Joe, Jerry, Duane, Vernon, Micheal, Greg, Frank, Bradley, Corey, Rodney, Angel, Derrick, Terrence

**MIDDLE_NAMES**

Anthony, Marcus, Jose, Kenneth, Lee, Colin, Arthur, Kirk, Blake, Dan, Benjamin, Marvin, Troy, Philip, Donald, Jamie, Calvin, Luke, Dustin, Marc, Tristan, Andres, Michael, Tyrone, Jeffery, Patrick, Wyatt, Luis, Larry, Frederick, Earl, Darrell, Perry, Roberto, Shannon, Douglas, Eddie, Jaime, Chad, Scott, Norman, Francis, Johnny, Ruben, Bernard, Albert, Rickey, Miguel, Spencer, Brent, Reginald, Leonard, Dennis, Kerry, Ronald, Russell, Gregg, Trevor, Drew, Hunter, Erik, Warren, Jesse, Levi, Francisco, Maxwell, Wayne, Ray, Lonnie, Ricky, Brian, Charles, Parker, Bryce, Bruce, Matthew, Clifford, Edwin, Nathan, Dean, Gordon, Sean, Stanley, Stephen, Karl, Dwayne, Antonio, Brady, Jeffrey, Elijah, Andrew, Adrian, Gilbert, Omar, Taylor, Tanner, Nathaniel, Devin, Lance, Harry

**LAST_NAMES**

Allison, Hanna, Stark, Mata, Travis, Peters, Zuniga, Smith, Gay, Thornton, Yu, Miller, Webb, Patterson, Ortiz, Combs, Meadows, Christensen, Freeman, Howell, Berger, Cooley, Glover, Jennings, Blackwell, Turner, Mcgee, Duffy, Montgomery, Glenn, Krause, Coleman, Petersen, Gregory, Barnes, Morris, Hensley, Harding, Bird, Estrada, Garza, Gomez, Burke, Waters, Lam, Davenport, Frost, Stafford, Jarvis, Williams,

**Dataset augmentation.**   For each family document, we create augmented training instances by randomly permuting the order of sentences (facts) while keeping each sentence unchanged. We generate 1,000 permutations per family, resulting in 318M tokens per training epoch.

## C    SVD-BASED UPDATE OF RELATION MATRICES IN RESCAL

In the RESCAL model (Nickel et al., 2011), the Alternating Least Squares (ALS) procedure requires updating the relation matrices $\{\mathbf{M}_r\}_{r=1}^m$ while holding the entity embedding matrix $\mathbf{A} \in \mathbb{R}^{n \times d}$ fixed, where $n$ is the number of entities, $d$ is the embedding dimension, and $m$ is the number of relations.

The objective function for a single relation $r$ is:

$$L(\mathbf{M}_r) = \frac{1}{2}\|\mathcal{X}_r - \mathbf{A}\mathbf{M}_r\mathbf{A}^T\|_F^2 + \frac{\lambda_R}{2}\|\mathbf{M}_r\|_F^2 \tag{6}$$

Our goal is to find the $\mathbf{M}_r \in \mathbb{R}^{d \times d}$ that minimizes this function.

To find the minimum, we can set the gradient of $L(\mathbf{M}_r)$ with respect to $\mathbf{M}_r$ to zero:

$$\frac{\partial L}{\partial \mathbf{M}_r} = -\mathbf{A}^T(\mathcal{X}_r - \mathbf{A}\mathbf{M}_r\mathbf{A}^T)\mathbf{A} + \lambda_R\mathbf{M}_r = 0 \tag{7}$$

Rearranging the terms, we get the normal equation:

$$\mathbf{A}^T\mathbf{A}\mathbf{M}_r\mathbf{A}^T\mathbf{A} + \lambda_R\mathbf{M}_r = \mathbf{A}^T\mathcal{X}_r\mathbf{A} \tag{8}$$

This is a continuous Sylvester equation. Using the vectorization operator vec() and the Kronecker product $\otimes$, we can rewrite it as a standard linear system:

$$\left((\mathbf{A}^T\mathbf{A}) \otimes (\mathbf{A}^T\mathbf{A}) + \lambda_R\mathbf{I}_{d^2}\right)\mathrm{vec}(\mathbf{M}_r) = \mathrm{vec}(\mathbf{A}^T\mathcal{X}_r\mathbf{A}) \tag{9}$$

Solving this equation directly requires inverting a dense $(d^2 \times d^2)$ matrix, which is computationally expensive with a complexity of $\mathcal{O}((d^2)^3) = \mathcal{O}(d^6)$. This becomes prohibitive as the embedding dimension $d$ grows. Due to high dimension of embedding space ($d = 896$) of our models, Eq. 9 is infeasible to solve directly.

To mitigate this issue, we employ the Singular Value Decomposition (SVD) to overcome the aforementioned computational bottleneck. Let the SVD of the entity matrix be $\mathbf{A} = \mathbf{U}\mathbf{S}\mathbf{V}^\top$ with orthonormal $\mathbf{U}, \mathbf{V}$ and singular values $\mathbf{S} = \mathrm{diag}(s_1, \ldots, s_d)$. Using $\mathbf{A}^\top\mathbf{A} = \mathbf{V}\mathbf{S}^2\mathbf{V}^\top$, Eq. 8 can be rotated into the singular space, yielding the diagonal Sylvester equation

$$s_i^2(\tilde{\mathbf{M}}_r)_{ij}s_j^2 + \lambda_R(\tilde{\mathbf{M}}_r)_{ij} = s_i(\tilde{\mathcal{X}}_r)_{ij}s_j, \qquad \tilde{\mathbf{M}}_r = \mathbf{V}^\top\mathbf{M}_r\mathbf{V}, \ \tilde{\mathcal{X}}_r = \mathbf{U}^\top\mathcal{X}_r\mathbf{U}.$$

Solving element-wise gives

$$(\tilde{\mathbf{M}}_r)_{ij} = \frac{s_is_j}{s_i^2s_j^2 + \lambda_R}(\tilde{\mathcal{X}}_r)_{ij},$$

or in matrix form $\tilde{\mathbf{M}}_r = \mathbf{P} \odot \tilde{\mathcal{X}}_r$ with $P_{ij} = s_is_j/(s_i^2s_j^2 + \lambda_R)$. Transforming back,

$$\boxed{\mathbf{M}_r = \mathbf{V}\left(\mathbf{P} \odot (\mathbf{U}^\top\mathcal{X}_r\mathbf{U})\right)\mathbf{V}^\top}.$$

The update costs $\mathcal{O}(nd^2)$ for the SVD and $\mathcal{O}(n^2d + d^3)$ for the remaining multiplications, vs. $\mathcal{O}(d^6)$ for the naive Kronecker inversion. In this work, we employed the algorithm in Nickel (2013) and reproduce the formula here only to motivate our implementation choice.

$n = 1250$ (max entities; 125 families, 10 members per family) and $d = 896$ (embedding dimension) in our experiments, making the SVD-based update feasible. $M_r$ is obtained from train set $A$ then evaluated on test set $B$. $B$ and $A$ have disjoint entity sets.

# D TRAINING, PROBING, AND EDITING RESULTS

## D.1 TRAINING RESULT IN DETAIL

In this section, we provide training curves for all models with different weight decay values and random seeds over training steps (see Fig 6). All models achieve 100% training accuracy, but test accuracy varies significantly based on weight decay and random seed.

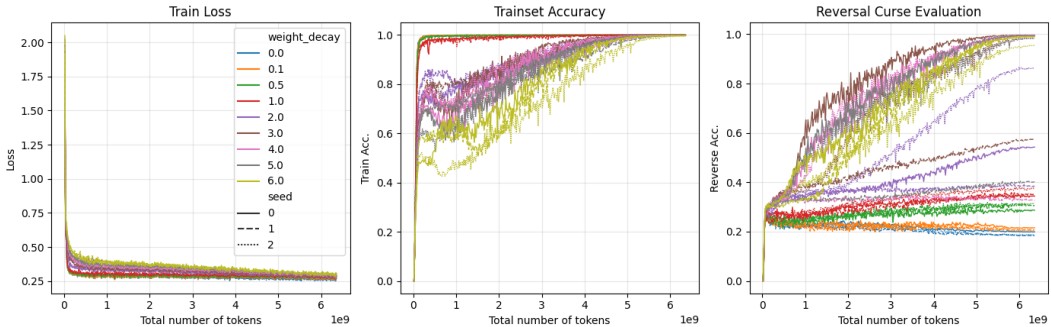

Figure 6: Training loss, train accuracy, and test accuracy for models with different weight decay values and seeds.

## D.2 PROBING RESULTS IN DETAIL

### D.2.1 LINEAR RELATION EMBEDDING

Figure 7 shows the probe's accuracy as a function of the number of training samples per relation, $n$ ($n = 10$, $n = 100$, and $n = 500$). The main text reports results for $n = 10$ due to the high computational cost of Jacobian calculations. Here, we show that increasing the number of samples up to $n = 500$ does not improve performance, confirming that the poor accuracy of the linear relational embedding probe is not due to insufficient sampling. We also run our bilinear probing for $n = 10$, $n = 100$, and $n = 500$, which shows that insufficient $n$ leads to underfitting.

Figure 8 visualizes the layer-wise accuracy for each of the eight relations individually for $N = 100$ and $\beta = 5$. Interestingly, few models in "Reversal cursed" group (blue) show high accuracy at mid-late layers while "Not Reversal Cursed" models (orange) do not. It indicates that some models in the "Reversal cursed" group do learn linear relational embeddings for certain relations, but they are not consistent across relations and layers.

### D.2.2 TRANSLATIONAL

Figure 9 shows the per-relation accuracy for this task. The "Not Reversal Cursed" models exhibit high accuracy only for the symmetric `husband` and `wife` relations, peaking at mid-to-late layers. Accuracy for all other relations is near zero for both model groups. This suggests that while a translational structure is learned, it is limited to simple symmetric pairs and does not generalize to other relation types.

### D.2.3 BILINEAR

Figure 10 shows the per-relation accuracy of bilinear probing. The "Not Reversal Cursed" models achieve high accuracy across all relations, peaking at mid-to-late layers. In contrast, "Reversal Cursed" models show near-zero accuracy for all relations. This indicates that learning a bilinear relational structure is strongly associated with overcoming the reversal curse and generalizes well across different relation types.

## D.3  EDITING RESULT IN DETAIL

**Setup.** For each model we sample 50 distinct husband facts (A, husband, B) from Group 1. Each is edited to (A, husband, B') where B' is another female entity from the *same* family (preserves name template and type). Single edit per run; no simultaneous multi-fact changes. For every layer $l \in \{1, \ldots, 12\}$ we fine-tune only the MLP output (final linear) weights of that layer using a single example $(A, \texttt{husband}, B')$ . Optimizer: Adam, lr $4 \times 10^{-4}$, early stop when loss $< 0.2$ (cap 50 update steps). All other parameters frozen.

**Metrics.** For each edited model:

- *Edit Success*: accuracy on (A, husband, B').
- *Logical Generalization (Reverse-relation)*: accuracy on (B', wife, A).
- *Logical Generalization (B', son/daughter, C/D)*: mean accuracy over (B', son/daughter, C/D).
- *Logical Generalization (C/D, father, B')*: mean accuracy over (C/D, father, B').
- *Locality (In-Family)*: accuracy on other facts inside the edited family excluding any incident to B'.
- *Locality (Other Families)*: accuracy on a fixed held-out set of facts from untouched families.

Accuracies are proportion of correct next-token generations for the object name (exact match). Curves in Figure 11 report the mean over the 50 independent edits.

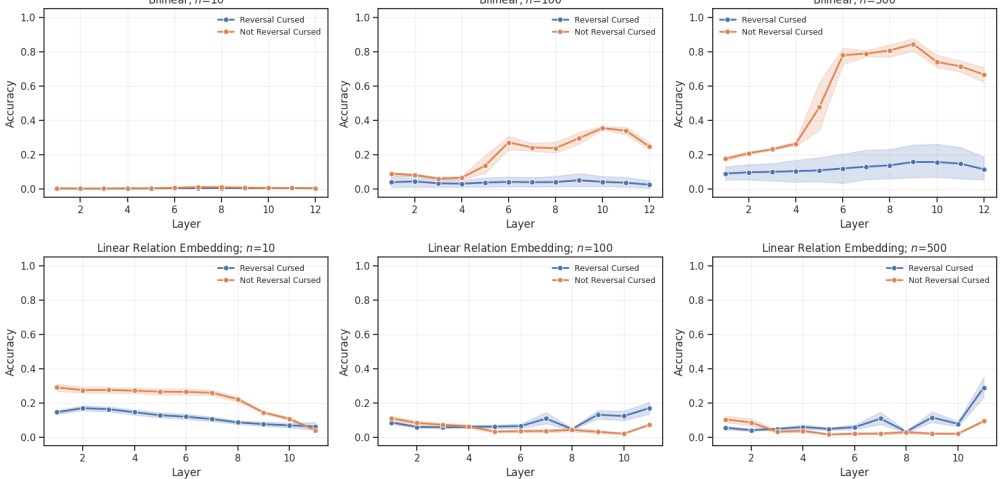

Figure 7: Bilinear and linear relational embedding accuracy for $n = 10$, $n = 100$, and $n = 500$, where $n$ denotes the number of training samples per relation.

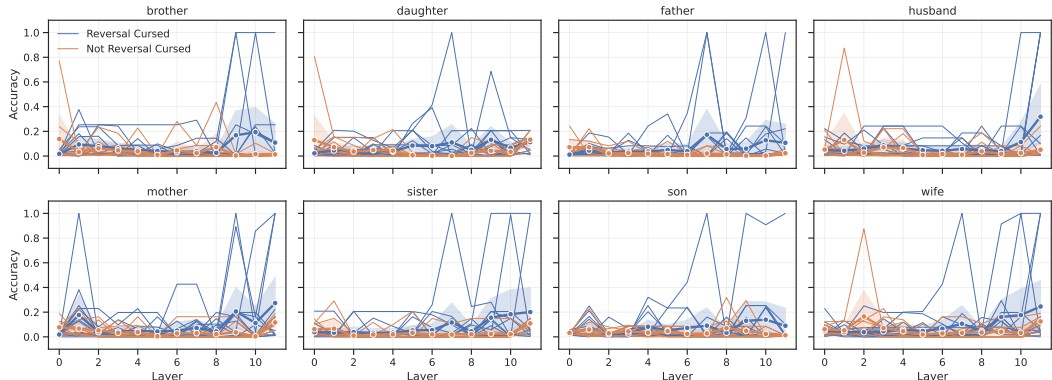

Figure 8: Visualization of linear relational embedding probing results for each relation $r$ as a spaghetti plot ($n = 100$ and $\beta = 5$).

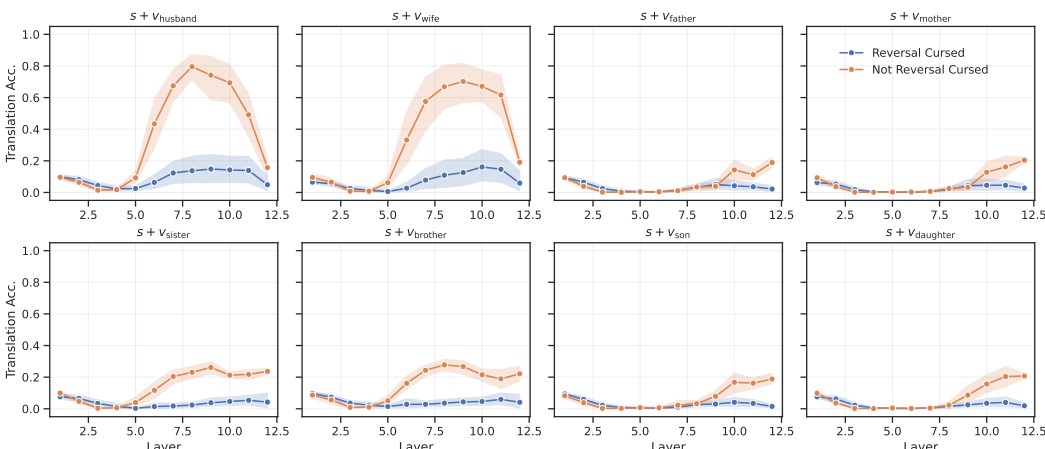

Figure 9: Translational probing accuracy for each relation $r$.

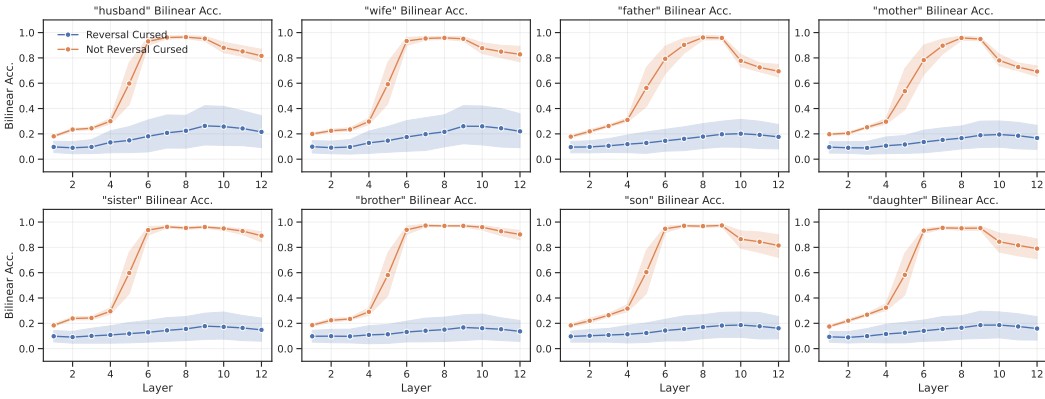

Figure 10: Bilinear probing accuracy for each relation $r$.

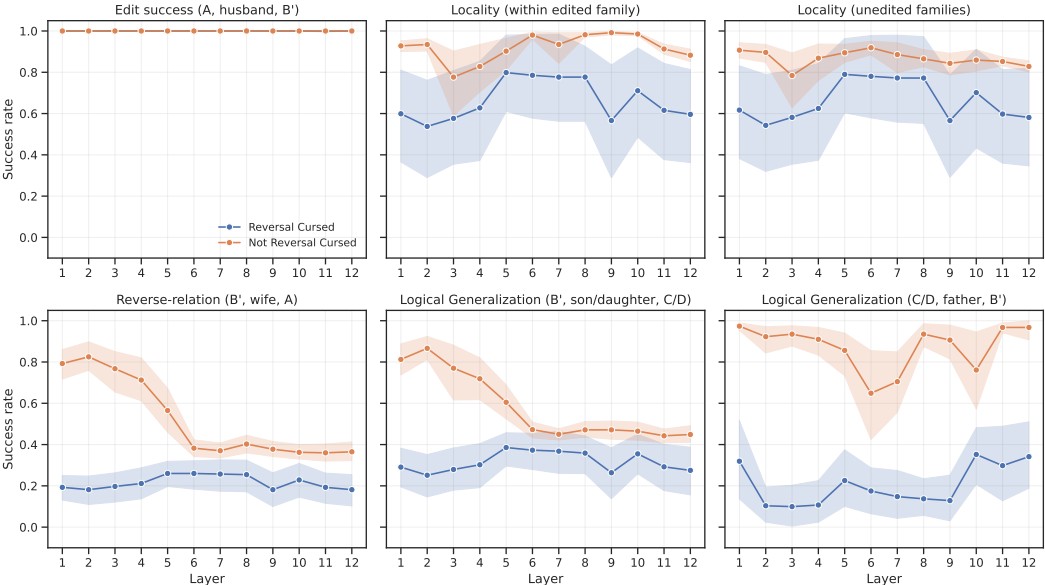

Figure 11: Editing experiment details. Six panels: Edit Success, Locality (edited family), Locality (other families), Reverse relation, Logical Generalization to children, Logical Generalization to parents.

