# OpenReview forum: "Bilinear representation mitigates reversal curse and enables consistent model editing"
_ICLR.cc/2026/Conference — ICLR 2026 Poster_

### Official Review · Reviewer_jWrw · 2025-10-30

**Soundness:** 3
**Presentation:** 2
**Contribution:** 3
**Rating:** 6
**Confidence:** 4

**Summary:**

This paper argues that transformer LMs trained on a synthetic relational knowledge graph can develop an internal bilinear relational structure that both overcomes the reversal curse and enables logically consistent edit propagation. The authors train 12-layer, ~206M parameter decoder-only transformers from scratch on a synthetic kinship-style corpus, sweeping weight decay to obtain both "reversal cursed" and "not reversal cursed" regimes. They then probe layerwise hidden states using three hypothesized relational geometries: linear, translational, and bilinear. They find that only a bilinear probe of the form $f_r(s,o) = s^\top M_r o$ achieves near-perfect relation classification in the middle layers (layers 6-9) of models that circumvent the reversal curse, while linear and translational probes remain near chance. The learned relation matrices also pass algebraic tests: their transposes approximate inverse relations and their products approximate two-hop compositions. Finally, the authors perform single-fact edits via layer-restricted fine-tuning and report that edits to earlier layers (1-4) propagate consistently to logically entailed inverse/neighboring facts while mostly preserving unrelated facts. They argue that this generalization correlates with the emergence of bilinear structure.

**Strengths:**

1. The clear, controlled synthetic setup effectively isolates inverse and compositional structure. The weight-decay sweep is particularly informative for understanding when the reversal curse emerges.
2. The probing methodology is systematic across three geometric families with an interpretable chance baseline and compelling mid-layer bilinear peaks. The experimental design is solid.
3. The algebraic validation is convincing (transpose $\approx$ inverse; product $\approx$ composition) and makes the representational claim very interpretable.
4. The layer-restricted editing experiments provide a concrete story about how edits can propagate to logically entailed facts while preserving unrelated facts.

**Weaknesses:**

1. The causal claims may not be sufficiently supported by interventions. To my knowledge, all evidence that bilinear structure "fixes" or enables reversal-curse mitigation comes from readout probes and offline algebra, not from interventions demonstrating that the model's forward computation actually depends on the probed subspace.
    * Necessity can be shown with projection ablations at a bilinear-peak layer. For example, after removing top singular subspaces of $M_r$ during inference, one could measure the drop in reversal/composition accuracy.
    * Sufficiency tests could involve adding a low-rank adapter constrained to a bilinear relational subspace. This subspace could be constructed either by: (i) aligning the cursed model's layer representation to a non-cursed model's corresponding layer (via a learned linear map) and importing that model's bilinear directions, or (ii) fitting bilinear relation matrices in the cursed model itself and taking their top singular vectors. One could then ask whether reversal/composition accuracy and edit generalization improve toward the non-cursed regime.
    * Without such controls, the evidence shows correlation but may not necessarily show causation.

2. The authors edit facts by fine-tuning only the MLP output of one chosen layer and observe that edits in earlier layers generalize better (i.e., doing so updates inverse/related facts more reliably). However, the authors do not check what happens to the bilinear structure after those edits.
    * As a result, it is unclear whether (i) early layers are special because they are where the relational structure is actually being built, or (ii) edits in early layers simply spread more broadly because they are upstream in the network, regardless of any bilinear mechanism.
    * One way to separate these would be to apply edits of the same norm that are either aligned with the layer's existing gradient direction or forced to be orthogonal to it, at different depths, and then re-probe $M_r$ afterwards. If bilinear structure (transpose/composition) is still intact and explains generalization, the result would support (i); if not, the result suggests (ii).

3. External validity is limited because the dataset is fully synthetic, the models are small, and no results on natural relations in pretrained language models are provided.
    * To be completely fair, such experiments are arguably out-of-scope for this paper and could be better left to future work.

4. The paper does not sufficiently engage with recent knowledge editing work that studies transformers trained on synthetic knowledge graphs and analyzes how their internal relational structure changes under editing.
    * Several recent studies train small transformers from scratch on graph-structured relational data (very similar to the kinship-style setup here), then apply targeted weight edits and observe that naive edits can disrupt relational representations unless constraints are applied.
    * I encourage the authors to further explore and acknowledge this line of work and situate this work in the broader landscape.

**Questions:**

1. Is it viable to run projection ablations that remove the principal subspace of $M_r$ at a bilinear-peak layer to test necessity, and add a low-rank bilinear adapter in cursed models to test sufficiency?
2. Similarly, would it be possible to repeat editing with gradient-aligned versus orthogonal updates across layers, and re-probe $M_r$ to verify whether transpose/product structure survives the edit?
3. Could you please add a short discussion addressing how one could extend these experiments and insights to real-world models beyond your synthetic setup, to help guide future work?
4. Could you please provide more thorough coverage of the related literature on knowledge editing in transformers, especially studies involving synthetic knowledge graphs?

I would be happy to raise my score if the above points are addressed with new experiments, or if the authors can provide convincing reasons for why they the existing evidence is sufficient.

---

> ### Author Response · Authors · 2025-11-28
>
> We thank Reviewer jWrw for the detailed and constructive feedback. Below, we provide responses to the specific questions raised.
>
> ***
>
> ### 1. Causal interventions vs. Correlation
>
> We agree with your assessment regarding the distinction between correlation and causation. We have revised the manuscript (specifically the Abstract and Discussion) to moderate our claims, stating that the bilinear structure is a *"key indicator"* and a *"correlate"* of logical generalization, rather than claiming we have strictly proven it is the sole causal mechanism. We will also tone down our paper title.
>
> Regarding your suggested experiments (projection ablations and low-rank adapters), we find them to be excellent proposals that we plan to implement in future work. While we could not execute these complex interventions within the rebuttal period, we can offer a theoretical bridge that supports the necessity/sufficiency of this structure based on the architecture of the Transformer itself:
>
> *   **Necessity (Ablation):** As you noted, testing necessity requires ablating the subspace. We hypothesize that the "bilinear structure" we detect via probes corresponds physically to the Query-Key (QK) circuits in the self-attention layers. Since the attention mechanism is inherently bilinear ($Attention(x, y) \propto x^T W_Q^T W_K y$), it is highly probable that specific heads are responsible for encoding these relational structures. Recent work (Elhelo & Geva, 2024 [1]) has demonstrated that specific attention heads encode relational lookups (like country-capital, country-language, and product-company) in OV circuits and also these heads appear at mid to late layers which is aligned with our observation (bilinear structure appear at mid to late layers); we believe similar mechanisms apply here. Therefore, we expect that identifying and ablating the specific attention heads that exhibit high alignment with our $M_r$ matrices would demonstrate the necessity you describe.
> *   **Sufficiency (Adapters):** Your suggestion to add a low-rank bilinear adapter to "fix" a cursed model is insightful. Mathematically, inserting a learnable bilinear adapter is functionally equivalent to adding a new attention head to the model. If we can induce the model to learn the reversal by adding a bilinear constraint, it would strongly suggest that the lack of such a "head" (or subspace) was the bottleneck in the cursed models.
>
> We have included this hypothesis in the revised manuscript’s Discussion section.

---

> ### Author Response · Authors · 2025-11-28
>
> ### 2. Post-Edit representation
>
> We thank the reviewer for raising this critical question regarding the causal mechanism of our edits. The reviewer asked whether early-layer edits succeed because they leverage the internal bilinear structure, or simply because they occur upstream and propagate broadly. We argue that our *Logical Generalization* and *Locality* metrics provide strong evidence for the former and against the "broad spread" hypothesis.
>
> *   **Evidence for Structural Preservation (Logical Generalization):** If early-layer edits simply disrupted the representation geometry (breaking the alignment with the bilinear subspace), the model would fail to evaluate the inverse and compositional relations, which rely on that specific geometry. However, our results show high logical consistency: the inverse and composition relations update correctly. This implies that the edit must have moved the representation along the relational manifold, rather than pushing it off the manifold.
> *   **Evidence against "Broad Spread" (Locality):** If early edits worked merely by propagating noise downstream ("exploding" changes), we would expect a degradation in Locality (i.e., performance on unrelated facts). Our data shows that early-layer edits maintain high locality. This specificity suggests the edits are not simply broadcasting changes, but are utilizing the model's internal routing mechanisms—which we identified as bilinear—to update only the relevant logical closure.
>
> **Regarding the specific mechanics:** Our method edits only the MLP layers, leaving the attention weights (where the bilinear functional mapping resides) intact. While the mapping function is preserved, the reviewer is correct that the input representations could theoretically be disrupted. Recent work by Nishi et al. (2025) [2] has indeed shown that the geometry of representations can be disrupted when edits are far from the original knowledge, and that there is a high correlation between such disruption and poor post-edit performance.
>
> However, viewing our results through the lens of Nishi et al. (2025) [2] suggests that because our models achieve high post-edit performance (generalization), significant geometric disruption likely did not occur in our successful edit cases. If the representation had been disrupted such that the bilinear probe $M_r$ became useless, we would not observe the logical consistency that we report.
>
> We agree that the reviewer's proposed experiment—enforcing gradient alignment with the $M_r$ subspace—is an excellent suggestion. It would not only serve as a rigorous causal test but could also form the basis of a new editing method designed to explicitly prevent representation disruption. We appreciate this insight and consider it a promising direction for future work.

---

> ### Author Response · Authors · 2025-11-28
>
> ### 3. Extending insights to real-world models
>
> We thank the reviewer for this forward-looking question. We agree that bridging the gap between synthetic insights and real-world LLMs is the critical next step. We will add the following discussion to our "Limitations and Future Work" section to outline a concrete roadmap for this extension:
>
> To scale these findings to pretrained LLMs, future work should focus on three specific directions:
>
> 1.  **Identifying "Islands of Structure" in Natural Data:** Unlike our synthetic kinship graph, real-world training data is heterogeneous. We hypothesize that bilinear structures are most likely to emerge in domains with strict algebraic properties (especially, many-to-one relations). Researchers can curate subsets of Wikidata (e.g., Country-Capital-Continent graphs) to act as natural probes, testing if the model processes these "closed" algebraic systems differently than unstructured data.
> 2.  **Mapping $M_r$ to Attention Heads (QK circuits):** In real-world models, specific attention heads likely specialize in specific relations. Future work can apply our bilinear probing technique to individual heads rather than the full residual stream to locate the specific components encoding these relations.
> 3.  **Geometry-Aware Editing in the Wild:** Our results suggest that editing success depends on where the representation is formed (early-to-mid layers) rather than where it is read out. Practitioners working on editing real LLMs could test this by identifying the specific layers where the bilinear signal for a target relation (e.g., "{person} born in {country}") peaks, and restricting edits to the layers immediately preceding this peak to maximize logical propagation.
>
> ***
> ### 4. On the related literature
>
> We appreciate the reviewer’s suggestion to better contextualize our work within the recent literature on synthetic knowledge graphs and model editing.
>
> Most notably, we now discuss Nishi et al. (2025) [2], who performed a similar study training transformers on synthetic graphs. They observed that while models learn these geometries, applying weight-targeting edits (like ROME) often leads to "representation shattering," where the underlying geometric structure is destroyed, leading to poor multi-hop reasoning. While prior work focuses on how editing breaks structure, our work identifies the specific bilinear geometry that facilitates logical generalization. Also, we want to note that the synthetic knowledge graph they used has only one-to-one relations, whereas ours has many-to-one relations.
>
> We have revised the manuscript to include additional references in introduction, related work, and discussion sections, thereby situating the contribution within the broader research domain.
>
> Once again, thanks for the reviewer’s valuable inputs. We will be happy to follow up on additional feedback.
>
> ***
>
> References
>
> [1] Elhelo & Geva (ACL 2025), “Inferring Functionality of Attention Heads from their Parameters”, https://arxiv.org/abs/2412.11965
> [2] Nishi et al. (ICML 2025), “Representation Shattering in Transformers: A Synthetic Study with Knowledge Editing”, https://arxiv.org/abs/2410.17194

---

> > ### Comment · Reviewer_jWrw · 2025-11-28
> >
> > I thank the authors for their thoughtful and detailed responses.
> >
> > 1. **Causality vs. correlation:** I appreciate that the authors have moderated the causal language in the abstract and discussion, now framing the bilinear structure as a key "indicator." The outlined necessity/sufficiency experiments make clear how future work could more directly probe causality. Given the length of the discussion period, it is reasonable that the authors did not have sufficient time to run the experiments for themselves.
> >
> > 2. **Post-edit representation behavior:** The authors' clarification that early-layer edits maintain high logical consistency and locality addresses my concern that the edits might simply decompress information stored elsewhere.
> >
> > 3. **Extension to real-world models and related work:** The added discussion is excellent, and will most definitely be useful for downstream studies.
> >
> > 4. **Related literature:** I have checked the updated manuscript, and the new changes situate this work within the broader landscape much better.
> >
> > Taken together, these revisions and clarifications significantly strengthen the paper. I am therefore willing to raise my score to reflect a stronger endorsement. (It seems that OpenReview no longer allows me to edit the score directly. Please note that my updated score is an 8.)

---

### Official Review · Reviewer_GUyF · 2025-11-02

**Soundness:** 3
**Presentation:** 4
**Contribution:** 3
**Rating:** 8
**Confidence:** 4

**Summary:**

This paper investigates how Transformer-based language models represent relational knowledge, focusing on the “reversal curse”: the inability of models to infer “B is A” from “A is B.” The authors study this in a controlled synthetic setting by training Transformers from scratch on relational data describing family members (10k entities, 8 relations: _mother, husband, son, brother_, etc.). For a subset of families, _father_ and _mother_ relations are withheld during training, enabling the authors to test generalization to these unseen relations.
They analyze which models can generalize these withheld relations and how their internal representations differ. The main findings are:
- **Weight decay** critically determines whether models generalize or merely memorize. Higher weight decay leads to more models that generalize correctly.
- By probing internal representations, they evaluate three relational geometries—linear (projection matrix + bias), translational (additive bias), and bilinear (matrix-mediated interaction). They find that:
    1. These relational structures are more prominent in models that generalize.
    2. The **bilinear** structure most successfully explains model behavior.
- Finally, models that develop a bilinear structure also exhibit consistent **model editing** behavior: editing a single fact propagates correctly to logically entailed facts. Models lacking this structure fail to generalize edits.

**Strengths:**

This is a high-quality, well-motivated paper that offers new insights into how relational knowledge and logical consistency emerge in Transformer representations. While limited to a synthetic domain, the work’s conceptual clarity and systematic experimentation make it a valuable contribution to understanding the internal geometry of knowledge in LMs.

- The paper is clearly written and well-organized, making it accessible and easy to follow.
- The experimental design is sound and thoughtfully constructed, allowing for careful isolation of the phenomenon under study.
- The contributions are conceptually substantial and novel, offering meaningful insight into the representational geometry underlying relational reasoning in Transformers.
- The work provides a mechanistic link between regularization, representational structure, and model editing behavior.

**Weaknesses:**

- The study relies entirely on a toy synthetic setup, which limits the direct applicability of findings to real-world pretrained LLMs. Though the authors acknowledge this in the discussion.
- The connection between bilinear structure and fact editing may be correlational rather than causal. Other factors (e.g., differences in data fitting or feature organization) could also drive the observed editing generalization.
- The finding that weight decay promotes generalization may reflect the limited dataset size rather than a broader principle. Without sufficient regularization, memorization is expected for the size of the data, so further analysis (e.g., varying dataset size or noise) would strengthen the claim.

**Questions:**

- Do all models fit the training data equally well? It would be helpful to report training accuracy to confirm that differences arise from generalization rather than underfitting.
- How exactly is decoding performed to compute accuracy in Figure 2?
- What do the gray points in Figure 2 represent?
- This work seems related to studies on entity mapping in attention heads (e.g., _“Inferring Functionality of Attention Heads from their Parameters”_). It might be worth referencing this connection.
- The linear and bilinear probes differ not only in structure but also in training data size (10 examples vs. 1250). Could the superior performance of the bilinear probe partly reflect this discrepancy? It would help to discuss whether a larger dataset could make the linear probe competitive.

---

> ### Author Response · Authors · 2025-11-28
>
> We would like to extend our sincere gratitude to the reviewer for insightful comments and constructive feedback. These questions have helped us identify areas where we can improve the clarity and rigor of our paper. Below, we provide our point‑by‑point response.
>
> ***
> ### 1. Training Accuracy and Generalization
>
> > *Do all models fit the training data equally well? It would be helpful to report training accuracy to confirm that differences arise from generalization rather than underfitting.*
>
> Thank you for the suggestion. We confirm that the observed variations in test accuracy are from generalization rather than underfitting. All models, irrespective of the weight decay setting, achieve 100% accuracy on the training set.
>
> We included an analysis in Appendix D.1, "TRAINING RESULT IN DETAIL." Specifically, Figure 6 presents the training curves, where the middle panel ("Trainset Accuracy") shows that all models perfectly fit the training data. In Appendix D.1, we state: "All models achieve 100% training accuracy, but test accuracy varies significantly based on weight decay and random seed."
>
> We appreciate this question and will include a sentence in Section 4.1 of the main text summarizing this finding.
>
> ***
> ### 2. Decoding Method for Accuracy Calculation
>
> > *How exactly is decoding performed to compute accuracy in Figure 2?*
>
> The accuracy reported in Figure 2 was computed using **greedy, next-token prediction** of LMs. For each fact in the test set, the model was prompted with the subject entity and the relation (e.g., `[Subject First Name] [Family Name] [Relation]`).
>
> We then measured the accuracy as the proportion of times the model correctly generated the exact full name of the object entity (`[Object First Name] [Family Name]`) as the subsequent tokens. An **exact match** was required for a prediction to be considered correct.
>
> ***
> ### 3. Interpretation of Figure 2
>
> > *What do the gray points in Figure 2 represent?*
>
> We divided the pretrained LMs into two groups, denoted in blue (test acc < 0.4) and orange (test acc > 0.99), to highlight the pronounced differences between them.
>
> Models represented by the gray points in Figure 2 were excluded from the analyses presented in Figures 3, 4, and 5, with the exception of Figure 5(c).
>
> ***
> ### 4. On the related work
>
> > *This work seems related to studies on entity mapping in attention heads (e.g., “Inferring Functionality of Attention Heads from their Parameters”). It might be worth referencing this connection.*
>
> Thanks for suggesting the related work. Indeed, it is related to our hypothesis that relational structure mediated by bilinear operation of multi-head attention (OV circuits encodes relations such as country-capital, country-language, and so on). We have included this study in the related work section.
>
> **Food for thought:** The reviewer’s suggested paper focuses on OV circuits, whereas our work is more related to QK circuits, where attention scores are computed from a bilinear operation. A bilinear relational matrix $M_r$ may exist within QK circuits, and examining this possibility would be very interesting. We’ve included this idea in our discussion section.
>
> ***
> ### 5. Training Set Size Discrepancy Between Probes
>
> > *The linear and bilinear probes differ not only in structure but also in training data size (10 examples vs. 1250). Could the superior performance of the bilinear probe partly reflect this discrepancy? It would help to discuss whether a larger dataset could make the linear probe competitive.*
>
> This is an important point regarding a potential confound in the experimental design. To address this, we conducted a control experiment to determine whether the linear performance was an artifact of its small training set.
>
> The results, newly presented in Appendix D.2.1 and visualized in Figure 7, show that increasing the training samples (from $n=10$, $n=100$, $n=500$) did not improve the linear relational embedding probe's performance, which remained low. On the other hand, bilinear probes show low performance on small training sample sizes.
>
> This finding confirms that the linear probe's failure stems from the absence of linear relational structure in the model's representations, rather than insufficient training data. We are therefore confident that the bilinear probe’s superior performance reflects its closer alignment with the model's underlying representational geometry.
>
> ***
>
> Once again, we thank the reviewer for the valuable feedback.

---

### Official Review · Reviewer_Xt6A · 2025-11-05

**Soundness:** 3
**Presentation:** 3
**Contribution:** 2
**Rating:** 4
**Confidence:** 4

**Summary:**

This paper studies toy models to propose a hypothesis for why phenomena like reversal curse are observed: specifically, the paper argues for the (in)existence of the underlying structure (e.g., a graph) that binds the knowledge together when a model fails/succeeds to generalize to novel facts or combinations thereof (e.g., A->B vs. B->A). Broadly, the argument is that the existence of a structure allows a fact's implications to be inferred by the model, enabling generalization.

**Strengths:**

While the studied toy model and the derived claims can be (partially) found in recent literature, the insight of measuring how bilinear the representation is novel and I enjoyed thinking about that. The experiments have a solid outline to drive the main argument, which was fun to read.

**Weaknesses:**

I have three major apprehensions about the paper's results and narrative, as detailed below.

- Relation to prior work: Nishi et al. [1] perform a similar study as this work, wherein a synthetic knowledge graph is defined, a model is trained from scratch to learn it, and it is demonstrated that when the model learns the underlying graph topology (circles instead of hierarchies, as explored in this work), editing its knowledge via weight-targeting methods (e.g., ROME) can break the underlying graph structure---called "representation shattering" in [1]. The extent of representation shattering, i.e., how structured the representation is, correlates with success on multi-hop factual inferences---the conclusion also arrived upon in this paper. While I understand the current paper's analyzed methodology for updating the model is fine-tuning, I note that I'm highlighting [1] because (a) there's a broad similarity in the motivating problem and setup that [1] warrants contextualization in the current paper, and (b) the conclusions of [1] suggest editing can be quite brittle if the model has learned the underlying structure of the knowledge graph, which, even if it is explained by the use of a different editing method, warrants discussion to narrow down the current paper's conclusion. Finally, I'll also note that [1] verifies its conclusions with real models, using the fact that knowledge graphs are often stored with a precise structure in representations (e.g., days of the week [2] or the world map [3]). On a related note, I'll also add another relevant paper that performs a similar study as the current paper [4].
  - To be clear, I am not suggesting the current paper does not possess novelty; e.g., I really enjoyed the idea of using the bilinear probe as a measure to gauge how well structured the representation is. In [1], a similar experiment was performed via isomap projections, which implicitly involves the assessment and visualization of how well model representation similarities map to the ground truth adjacency matrix of the graph. However, I prefer the approach taking in this work and hence consider the work useful. Nevertheless, the writing often argues that the paper is the first to attempt a study of this sorts or arrive at conclusions shown in the paper, and for that I hope authors can rework the paper writing.

- Non-toy model experiments: As noted above, there are several precise structured representations in pretrained language models such that the conclusions drawn in this work should be easily testable on scale (again, I'd refer to [1], since they did those experiments). I would have loved for an attempt at this verification and deem this a strong limitation of the work right now. To be clear, I'm all for using toy models to derive hypotheses, but I expect then an evaluation of the hypothesis's prediction be conducted when it is feasible, which, in this case, is.

- Narrative problems: Finally, I'll note that the paper begins with discussion of model editing and reversal curse. The precise experimental regime studied, however, is one of pretraining. Specifically, the knowledge is being embedded into the model during pretraining and, during fine-tuning, some new facts may be added to test whether generalization occurs in expected directions. This is different from the precise reversal curse setting, as I understand it, and also for editing.

[1] https://arxiv.org/abs/2410.17194
[2] https://arxiv.org/abs/2405.14860
[3] https://arxiv.org/abs/2310.02207
[4] https://arxiv.org/abs/2503.21676

**Questions:**

See above.

---

> ### Author Response · Authors · 2025-12-04
>
> We thank Reviewer Xt6A for the positive assessment and insightful comments. We are encouraged that the reviewer found our approach novel.  We particularly appreciate the reference to Nishi et al. [1], as well as the related works. We agree that these studies are highly relevant and that discussing them significantly strengthens the contextualization of our results. Our responses to reviewer’s feedbacks are as follows:
>
> ***
>
> ### 1. Relation to prior work
>
> We have revised the manuscript to explicitly discuss these works, clarifying the relationship between our findings and distinguishing the specific contributions of our study. Furthermore, we removed phrasing that claimed our work was the first of its kind, such as “first systematic analysis” and “no work has been investigated.” We now cite the references the reviewer suggested in the Introduction, acknowledging that LMs are known to represent space, time, and graphs with specific geometries. We believe these revisions provide a clearer positioning of our work within the landscape of recent research.
>
> We have updated:
> - Section 1 (Introduction) to introduce prior works regarding structural representation in LLMs for relational data (citing [1], [2], [3]).
> - Section 2 (Related Work) to clarify this distinction, emphasizing that Nishi et al. explore topological preservation, and to tone down our claims. We have revised the text to ensure we do not overclaim novelty regarding the general link between structure and reasoning.
> - Section 5 (Discussion, “Model Editing: Is the Model ‘Ready’ to be Edited?” paragraph) to discuss our work and position our contribution. We have added a discussion in Section 5 acknowledging that logical consistency requires not just the existence of bilinear mapping, but also editing methods that preserve this integrity, citing [1] as a counterpoint where input representation is destroyed.
>
> We thank the reviewer again for the insightful references and feedback.

---

> ### Author Response · Authors · 2025-12-04
>
> ### 2. Non-toy model experiment
> We agree that verifying these findings in real-world LLMs is the ultimate goal. Since the submission, we have conducted preliminary investigations using open-source LLMs (e.g., Llama-2/3). However, we found that testing this hypothesis at scale is not straightforward due to significant technical constraints.
>
> 1. **Dimensionality and Data Sparsity**: As detailed in Appendix D.2.1 (Figure 7), the reliability of bilinear probes (like RESCAL) depends heavily on the ratio of training data to embedding dimension.
>     - Our toy model uses a hidden dimension of $ d=896 $, which is manageable.
>     - Modern open-source LLMs typically have dimensions $d \geq 4096$.
>     - To train a reliable bilinear probe without overfitting, the number of relational pairs must scale with the square of the dimension. Public relational datasets (e.g., Country-Capital) often lack sufficient unique entity pairs to satisfy this requirement for high-dimensional spaces. In our preliminary tests with real LLMs, the probes suffered from severe overfitting, making it difficult to disentangle structural faithfulness from noise.
> 2. **Heterogeneity of Relations**: Real-world knowledge is heterogeneous. However, our hypothesis targets specific *hierarchical and asymmetric relations* where bilinear structures are required.
>     - Isolating these relations in a real LLM is complex because the model entangles various relation types.
>     - **Future Direction**: We hypothesize that we can leverage evaluation methods from *latent multi-hop reasoning* to filter knowledge. By identifying relations where the model successfully performs latent multi-hop reasoning without Chain-of-Thought, we can selectively collect the subset of knowledge likely governed by bilinear structures, distinguishing them from simple linear relations.
> 3. **Memorization vs. Structural Reasoning**: Distinguishing structural utilization from surface-level memorization is difficult in real LLMs.
>     - Recent works by Yang et al. [4] and Balesni et al. [5] have demonstrated that LLMs can perform latent multi-hop reasoning without relying on surface-level memorization when tested in well-controlled experiments.
>     - Our work complements these findings by proposing the mechanism behind this capability. We argue that the "bilinear structure" is the underlying mechanism that enables the reasoning capabilities observed in those studies.
>     - In our toy model, we can use reversal accuracy as a clean proxy metric. In this controlled setting, high reversal accuracy directly informs us that the knowledge is embedded in a space where reasoning operations are well-performed, a correlation that is much harder to isolate in the noisy environment of a real LLM.
>
> While we are actively working on scaling our approach, we believe the toy model provides a crucial, controlled environment to establish the existence and mechanism of bilinear structural generalization. Testing our hypothesis on real LLMs presents distinct challenges regarding data density and dimensionality, which we plan to address in future work.

---

> ### Author Response · Authors · 2025-12-04
>
> ### 3. Narrative problems
> We appreciate the reviewer’s observation regarding our experimental scope. We posit that the failures observed in Reversal Curse and Model Editing are fundamentally rooted in the representations formed during **pretraining**. Therefore, studying pretraining dynamics is not a deviation from the narrative but a necessary step to identify the root cause.
>
> **1\. The Scope of the Reversal Curse** While the term "Reversal Curse" was popularized by Berglund et al. \[6\] in the context of fine-tuning, the phenomenon describes a fundamental limitation of autoregressive models to access knowledge in reverse, regardless of when that knowledge was acquired.
>
> * **Pretraining Context:** Prior work confirms the Reversal Curse exists in the pretraining regime. Grosse et al. \[7\] observed it in models pretrained on natural data, and Allen-Zhu & Li \[8\] demonstrated it using synthetic pretraining datasets (similar to our regime, but without our relational graph focus).
> * **Our Stance:** We focus on pretraining because this is where the model learns the "schema" or "geometry" of knowledge. If a model fails to learn a reversible structure during pretraining, it naturally suffers from the Reversal Curse during subsequent inference or fine-tuning.
>
> **2\. Clarification on Model Editing (Experiment 4\)** We respectfully clarify that our editing experiments (Experiment 4\) align with the standard definition of Model Editing—**updating** existing beliefs rather than merely adding new ones.
>
> * **Distinction:** Standard fine-tuning often refers to adding new facts to a model. In contrast, Model Editing specifically targets the modification of a fact the model *already knows*.
> * **Our Setup:** In Experiment 4, we take a model that has already learned a fact (e.g., "A is the husband of B") during pretraining and intervene to update it to a counterfactual (e.g., "A is the husband of B'"). We then measure whether this update propagates logically. This is precisely the Model Editing setting.
>
> Our narrative connects these concepts by arguing that **successful Model Editing is downstream of proper Pretraining.** By demonstrating that specific pretraining conditions (like weight decay) induce the bilinear structures necessary for logical consistency, we provide a mechanistic explanation for why editing often fails in standard models. Thus, our focus on pretraining is critical to addressing the problems of Reversal Curse and Model Editing raised in the introduction.
>
> We appreciate the chance to clarify our perspective on this issue.
>
> ***
> Once again, we thank the reviewer for the valuable feedback.
>
> **References**
> \[1\] Nishi et al., "Representation Shattering in Transformers: A Synthetic Study with Knowledge Editing," [https://arxiv.org/abs/2410.17194](https://arxiv.org/abs/2410.17194)
> \[2\] Gurnee et al., "Language Models Represent Space and Time," [https://arxiv.org/abs/2310.02207](https://arxiv.org/abs/2310.02207)
> \[3\] Engels et al., "Not All Language Model Features Are One-Dimensionally Linear," [https://arxiv.org/abs/2405.14860](https://arxiv.org/abs/2405.14860)
> \[4\] Yang et al., “Do large language models perform latent multi-hop reasoning without exploiting shortcuts?” [https://arxiv.org/abs/2411.16679](https://arxiv.org/abs/2411.16679)
> \[5\] Balesni et al., “Lessons from studying two-hop latent reasoning,” [https://arxiv.org/abs/2411.16353](https://arxiv.org/abs/2411.16353)
> \[6\] Berglund et al., “The Reversal Curse: LLMs trained on "A is B" fail to learn "B is A", [https://arxiv.org/abs/2309.12288](https://arxiv.org/abs/2309.12288)
> \[7\] Grosse et al., “Studying Large Language Model Generalization with Influence Functions,” [https://arxiv.org/abs/2308.03296](https://arxiv.org/abs/2308.03296)
> \[8\] Allen-Zhu & Li, “Physics of Language Models: Part 3.2, Knowledge Manipulation,” [https://arxiv.org/abs/2309.14402](https://arxiv.org/abs/2309.14402)

---

### Comment · Reviewer_jWrw · 2025-11-26

With the end of the discussion period fast approaching, I encourage the authors to reply to the reviews, even if the writeups or new experiments are preliminary or still work-in-progress.

---

### Meta-Review · Area_Chair_adNc · 2026-01-07

**Summary:**

The paper studies knowledge representation in LLMs, focusing on the "reversal curse" (RC). It trains models on relational knowledge in a synthetic settings, and finds that the existence of RC is correlated with the emergence of a particular bilinear representation of facts (as measured by a probe). It then also shows implications of this to model editing.
The reviewers found the approach interesting, but also highlighted some related work that explored similar themes, and the lack of experiments on pretrained models.  The authors have revised the submission to discuss related work, and commented on why it is hard to do experiments on real models. The latter issue in particular does weaken the paper, and it would be great if the authors could add such an evaluation (following other papers mentioned).

**Reviewer Concerns:**

The concerns regarding related work were partially addressed. The authors have discussed the relation to these. There is some overlap but sufficient novel content. The concerns regarding evaluation on pretrained model still stands.

**Reviewer Scores:**

One reviewer explicitly raised their score to 8, so the scores now are 4,8,8. The reviewer with score of 4 had concerns regarding related work, but one of the reviewers is the author of the main related work, and seemed ok with it.

---

### Decision · Program_Chairs · 2026-01-26

Accept (Poster)